# OpenReview forum: "Controlling Output Rankings in Generative Engines for LLM-based Search"
_ICLR.cc/2026/Conference — Submitted to ICLR 2026_

### Official Review · Reviewer_8QWw · 2025-10-15

**Soundness:** 2
**Presentation:** 2
**Contribution:** 3
**Rating:** 4
**Confidence:** 3

**Summary:**

The paper introduces CORE, a method designed to manipulate product rankings in LLM-based search systems by appending content to target items. The authors point out that LLM-based search engines heavily depend on the initial retrieval order from search engines and address this through two optimization approaches: (1) a shadow-model solution that uses gradient-based optimization with a surrogate model to approximate the target LLM's ranking behavior, and (2) a query-based solution that employs an iterative generator-optimizer loop without requiring model internals access. To evaluate their method, the authors construct AmazonCOREBench and show experiments on four commercial LLMs to show that CORE achieves promotional success.

**Strengths:**

1. The paper addresses an important problem in modern LLM-based search systems with a realistic black-box threat model that assumes no control over LLM architecture or search engine selection, only modifiable item metadata. The construction of AmazonCOREBench with 15 categories and 3,000 products provides a substantial, domain-relevant benchmark.
2. The evaluation is thorough, testing across four major commercial LLMs with multiple metrics including Promotion Success Rate, perplexity-based fluency assessment, and human evaluation studies with 20 annotators. The ablation studies systematically examine shadow model choices and other factors.
3. Especially, the query-based solution represents a significant technical contribution by reformulating gradient-free optimization as an iterative generator-optimizer loop that operates entirely through black-box LLM interactions.

**Weaknesses:**

1. While the paper formulates ranking control as minimizing the loss function in eq. 2, there is no explanation/analysis of why this objective leads to effective ranking manipulation. Also, there is no explanation of how the discrete reconstruction step is performed and how it affects optimization quality.
2. While Table 10 shows similarity scores of 3.7-4.7 between the shadow model and target LLMs on 10 samples, this validation is limited in scope. The paper does not analyze when the shadow model fails to approximate target LLM behavior, what product types or query styles cause divergence, or how approximation errors compound during 2000 gradient descent iterations.
3. The paper does not explore whether existing adversarial text detectors, consistency checks across multiple queries, or temporal pattern analysis could identify manipulated content. The ethical implications section is generic and does not propose concrete safeguards.
4. The transferability across categories and temporal robustness (as LLMs update) remain unexplored.

**Questions:**

Q0: Authors are requested to respond to the weaknesses highlighted in the above section.
Q1: The addition of Gaussian noise for exploration is mentioned but not systematically evaluated. In the experiments, how was the impact of adding this noise?
Q2: Have you observed cases where the optimization converges in embedding space but produces poor discrete text that fails to achieve desired rankings?
Q3: Given the work demonstrates significant ranking manipulation capabilities, what specific technical defenses do you recommend for LLM-based search systems? How might search platforms detect systematic ranking manipulation at scale, and what design changes to generative engines could increase robustness against optimization attacks like CORE while preserving utility for legitimate users?

---

> ### Author Response · Authors · 2025-11-25
>
> Thank you for your thoughtful and encouraging feedback. We are glad that you found our work to address an important problem and that you consider our evaluation thorough. We have carefully considered your concerns and respond to them individually below.
>
> **W1. While the paper formulates ranking control as minimizing the loss function in eq. 2, there is no explanation/analysis of why this objective leads to effective ranking manipulation. Also, there is no explanation of how the discrete reconstruction step is performed and how it affects optimization quality.**
>
> **RW1.** We thank the reviewer for the helpful question. The effectiveness of  Eq. 2 is because the LLM generates the ranked list in a fixed textual format (e.g., “Top-1: …”, “Top-2: …”), where each ranking position corresponds to a specific segment of the autoregressive decoding process. The probability term $p_\theta\big(z(T(i^\star)) \mid q, \mathcal{I}\big)$ therefore decomposes into the exact next-token probabilities the model uses when deciding which item to place in each slot. By minimizing the negative log-probability of a ranking where the target item appears earlier, the optimization increases the model’s likelihood of selecting that item during the early decoding steps that define higher-ranked positions, making Eq. 2 directly aligned with how the LLM produces the final ordering.
>
> Regarding the discrete reconstruction step, our approach follows standard embedding-based optimization: the target text is updated in the continuous embedding space and then projected back into discrete tokens using top-k nearest-neighbor decoding. This projection necessarily introduces a degree of semantic drift, since the discrete tokens cannot perfectly preserve the optimized embedding direction. To quantify this effect, we ran the shadow-model optimization on 500 randomly generated strings and compared the optimized continuous embedding with the embedding of the reconstructed discrete tokens. The results are shown below.
>
> | Metric            | Mean | Std  |
> |-------------------|------|------|
> | Cosine similarity | 0.81 | 0.06 |
>
> The cosine similarity of 0.81 indicates that the reconstructed text preserves most, but not all, of the optimized signal.
>
> We will clarify these points in the revised version.
>
> **W2.** While Table 10 shows similarity scores of 3.7-4.7 between the shadow model and target LLMs on 10 samples, this validation is limited in scope. The paper does not analyze when the shadow model fails to approximate target LLM behavior, what product types or query styles cause divergence, or how approximation errors compound during 2000 gradient descent iterations.
>
> **RW2.** We apologize for the confusion and thank the reviewer for raising this point. We clarify that Table 10 is not intended as a full validation of shadow-model fidelity, but rather as a sanity check that the shadow model mimics the general ranking tendencies of the synthesizing LLM. As described in Section 3.4.1, the shadow model is constructed via few-shot prompting using a small set of real (query, candidate set, ranked output) examples. No training or parameter updates are performed. The goal is therefore behavioral alignment, not a learned approximation whose errors accumulate over time.

---

> ### Author Response · Authors · 2025-11-25
>
> **W3. The paper does not analyze when the shadow model fails to approximate target LLM behavior.**
>
> **RW3.** Thank you for raising this point. The shadow model typically fails when it is too weak to approximate the target LLM via few-shot examples. Based on our assumption in line 215, we believe that few-shot examples can force the shadow model to mimic the ranking behavior of the synthesizing LLM. Accordingly, we have added the theoretical analysis for shadow–target mismatch.
>
> Assumption C.6 introduces a deviation parameter $\delta$ between the shadow model and the target LLM, and Theorem C.7 shows that the promotion probability on the target LLM is lower bounded by: $p_{\text{target}} \cdot \exp(-\delta)$.
>
> This provides a failure condition: when $\delta$ becomes large (i.e., the shadow model assigns content scores significantly different from the target LLM), the optimized text no longer transfers reliably, and the shadow model can be regarded as failing to approximate the target behavior. Appendix C.4 provides the full proof and explains how such mismatch propagates through the optimization trajectory.
>
> Empirically, Table 6 shows that when the shadow model is comparable in scale and capability (Llama-3.1-8B, Vicuna-13B, Mistral-7B), performance remains stable across models, with PSR differences within a small range (typically 3–6% between Llama-3.1-8B and Vicuna-13B), corresponding to cases where $\delta$ remains small.
>
>
> To illustrate the failure mode when the mismatch becomes large, we added a diagnostic experiment using a much weaker shadow model (Qwen2.5-0.5B-Instruct). We run the same Home & Kitchen experiment for GPT-4o, the results as shown below:
>
> | Model                     | Top-5          | Top-3          | Top-1          |
> |---------------------------|----------------|----------------|----------------|
> | Llama-3.1-8B              | 55.0%          | 45.5%          | 33.0%          |
> | Qwen2.5-0.5B-Instruct     | 37.0% (18%↓)   | 28.0% (17.5%↓) | 13.5% (19.5%↓) |
>
>
> These results empirically validate the theoretical analysis: when the shadow model is too weak to approximate the target LLM, optimization performance degrades significantly.
>
> **W4. What product types or query styles cause divergence, or how approximation errors compound during 2000 gradient descent iterations.**
>
> **RW4.** Regrading what product types or query styles cause divergence, or how approximation errors compound during 2000 gradient descent iterations. We added a theoretical analysis in Appendix C.2 showing that the ranking objective used in CORE exhibits piecewise-stable behavior with respect to the target item’s embedding, and that the optimization is tolerant to bounded proxy-model noise. Under this formulation, shadow-model approximation error affects only the gradient direction, and such noisy gradient updates are known to converge to stable regions without catastrophic error accumulation.
> Empirically, we observe no meaningful divergence across product categories or query types. Table 1 reports consistently high promotion success rates across all 15 categories (Top-3 PSR typically above 80%), and the variance across categories is small. This suggests that differences in product semantics (e.g., Electronics vs. Clothing) do not lead to systematic failure of the shadow-model optimization.
>
> In addition, regarding the reviewer’s concern about error compounding over 2000 gradient descent iterations, we evaluated the promotion success rate at different numbers of iterations (0, 500, 1000, 2000, 3000) on the Home & Kitchen. The results are shown below.
> | Iteration | **GPT-4o** |  |  | **Gemini-2.5** |  |  | **Claude-4** |  |  | **Grok-3** |  |  |
> |-----------|------------|----|----|----------------|----|----|----------------|----|----|--------------|----|----|
> |           | Top-5 | Top-3 | Top-1 | Top-5 | Top-3 | Top-1 | Top-5 | Top-3 | Top-1 | Top-5 | Top-3 | Top-1 |
> | 0     | 0.0%  | 0.0%  | 0.0%  | 0.0%  | 0.0%  | 0.0%  | 0.0%  | 0.0%  | 0.0%  | 0.0%  | 0.0%  | 0.0%  |
> | 500   | 51.0% | 42.0% | 30.0% | 49.5% | 40.5% | 29.0% | 55.0% | 45.0% | 33.0% | 48.5% | 39.5% | 28.5% |
> | 1000  | 53.0% | 44.0% | 31.5% | 51.5% | 42.0% | 30.0% | 57.0% | 46.0% | 34.0% | 50.0% | 41.0% | 29.5% |
> | 2000  | 55.0% | 45.5% | 33.0% | 53.5% | 43.0% | 31.0% | 56.5% | 46.5% | 33.0% | 51.5% | 42.0% | 30.5% |
> | 3000  | 56.0% | 46.5% | 33.5% | 54.0% | 43.0% | 31.5% | 60.5% | 48.5% | 35.0% | 52.0% | 42.5% | 30.5% |
>
> Performance grows steadily in the early iterations (0–1000) and largely stabilizes around 2000 iterations. At 3000 iterations, all models exhibit only small fluctuations with no sign of degradation. This indicates that approximation noise from the shadow model does not accumulate over long optimization horizons.
>
> We’ll add these experiments to the revised paper.

---

> > ### Author Response · Authors · 2025-11-25
> >
> > **W5. The paper does not explore whether existing adversarial text detectors, consistency checks across multiple queries, or temporal pattern analysis could identify manipulated content. The ethical implications section is generic and does not propose concrete safeguards./ Q3: What specific technical defenses do you recommend for LLM-based search systems?**
> >
> > **RW5/A3.** Inspired by your suggestion, we introduce three defense methods: (1) perplexity-based filtering, which removes items whose perplexity exceeds a fluency threshold; (2) pattern-based filtering, which filters out fields containing structured reasoning traces from a predefined pattern list; (3) length constraints, which discard content exceeding a maximum token budget.
> >
> > Specifically, we set the perplexity threshold to 50, based on the perplexity statistics reported in Table 3 (Section 4.4, Page 8); use a predefined pattern list consisting of ***["I'm exploring", "I'm analyzing", "I'm evaluating", "I'm examining", "I'm comparing", "I'm reviewing", "I'm explaining", "I'm outlining", "I'm breaking down", "I'm identifying", "I analyzed", "I evaluated", "I compared", "first", "next", "then", "finally", "step", "in conclusion", "this suggests"]***; and apply a 4000-word length constraint to filter unusually long fields. Note that Claude-3.7 has been deprecated, so we report results using Claude-4 instead. The results on Home & Kitchen are shown below.
> > | Defense | Method | **GPT-4o** |  |  | **Gemini-2.5** |  |  | **Claude-4** |  |  | **Grok-3** |  |  |
> > |---------|--------|------------|------|------|----------------|------|------|----------------|------|------|--------------|------|------|
> > |         |        | Top-5 | Top-3 | Top-1 | Top-5 | Top-3 | Top-1 | Top-5 | Top-3 | Top-1 | Top-5 | Top-3 | Top-1 |
> > | /       | String    | 55.0% | 45.5% | 33.0% | 53.5% | 43.0% | 31.0% | 56.5% | 46.5% | 33.0% | 51.5% | 42.0% | 30.5% |
> > |         | Reasoning | 92.0% | 87.0% | 81.0% | 88.5% | 84.0% | 77.5% | 88.5% | 84.0% | 76.5% | 88.0% | 83.0% | 77.0% |
> > |         | Review    | 89.5% | 85.0% | 79.0% | 94.5% | 89.5% | 83.5% | 91.5% | 87.0% | 81.0% | 90.5% | 85.0% | 78.5% |
> > | perplexity-based filtering | String | 0.0% (55.0%↓) | 0.0% (45.5%↓) | 0.0% (33.0%↓) | 0.0% (53.5%↓) | 0.0% (43.0%↓) | 0.0% (31.0%↓) | 0.0% (56.5%↓) | 0.0% (46.5%↓) | 0.0% (33.0%↓) | 0.0% (51.5%↓) | 0.0% (42.0%↓) | 0.0% (30.5%↓) |
> > |         | Reasoning | 12.0% (80.0%↓) | 8.0% (79.0%↓) | 3.5% (77.5%↓) | 10.0% (78.5%↓) | 7.0% (77.0%↓) | 3.0% (74.5%↓) | 13.0% (75.5%↓) | 8.5% (75.5%↓) | 3.0% (73.5%↓) | 11.0% (77.0%↓) | 7.5% (75.5%↓) | 3.0% (74.0%↓) |
> > |         | Review    | 84.0% (5.5%↓) | 80.0% (5.0%↓) | 72.5% (6.5%↓) | 89.0% (5.5%↓) | 84.0% (5.5%↓) | 76.0% (7.5%↓) | 90.0% (1.5%↓) | 86.0% (1.0%↓) | 78.0% (5.5%↓) | 85.0% (5.5%↓) | 80.5% (4.5%↓) | 72.0% (6.5%↓) |
> > | pattern-based filtering | String | 55.0% (0.0%↓) | 45.5% (0.0%↓) | 33.0% (0.0%↓) | 53.5% (0.0%↓) | 43.0% (0.0%↓) | 31.0% (0.0%↓) | 56.5% (0.0%↓) | 46.5% (0.0%↓) | 46.5% (0.0%↓) | 51.5% (0.0%↓) | 42.0% (0.0%↓) | 30.5% (0.0%↓) |
> > |         | Reasoning | 35.0% (57.0%↓) | 25.0% (62.0%↓) | 16.0% (65.0%↓) | 33.0% (55.5%↓) | 23.0% (61.0%↓) | 15.0% (62.5%↓) | 38.0% (50.5%↓) | 28.0% (56.0%↓) | 18.5% (57.5%↓) | 32.0% (56.0%↓) | 22.0% (61.0%↓) | 14.0% (63.0%↓) |
> > |         | Review    | 69.0% (20.5%↓) | 64.0% (21.0%↓) | 56.0% (23.0%↓) | 74.0% (20.5%↓) | 68.5% (21.0%↓) | 60.0% (23.5%↓) | 75.0% (16.5%↓) | 69.5% (17.5%↓) | 61.0% (21.0%↓) | 71.0% (19.5%↓) | 66.0% (19.0%↓) | 57.0% (21.5%↓) |
> > | length constraints | String | 55.0% (0.0%↓) | 45.5% (0.0%↓) | 33.0% (0.0%↓) | 53.5% (0.0%↓) | 43.0% (0.0%↓) | 31.0% (0.0%↓) | 56.5% (0.0%↓) | 46.5% (0.0%↓) | 46.5% (0.0%↓) | 34.0% (0.0%↓) | 42.0% (0.0%↓) | 30.5% (0.0%↓) |
> > |         | Reasoning | 52.0% (40.0%↓) | 44.0% (43.0%↓) | 31.0% (50.0%↓) | 49.0% (39.5%↓) | 41.0% (43.0%↓) | 29.0% (48.5%↓) | 54.0% (34.5%↓) | 45.0% (39.0%↓) | 33.0% (43.5%↓) | 50.0% (38.0%↓) | 41.0% (42.0%↓) | 29.0% (48.0%↓) |
> > |         | Review    | 76.0% (13.5%↓) | 70.0% (15.0%↓) | 62.0% (17.0%↓) | 81.0% (13.5%↓) | 75.0% (14.5%↓) | 66.0% (17.5%↓) | 82.0% (9.5%↓) | 76.0% (11.0%↓) | 68.0% (13.0%↓) | 78.0% (12.5%↓) | 72.5% (12.5%↓) | 64.0% (14.5%↓) |
> >
> > Perplexity-based filtering eliminates nearly all string-based content and reduces reasoning-based content by more than 75% across models, while preserving most review-style content. Under pattern-based filtering and length constraints, the success rates of both reasoning and review strategies also drop sharply, reflecting their strong dependence on structured phrasing and long-form text. Note that string-based content remains unchanged under these two defenses, as it is short and does not contain the reasoning patterns that trigger filtering, making these defenses not applicable to the string strategy. Review still remains the strongest overall performer under defense mechanisms, highlighting the need for more robust ranking-protection mechanisms.
> >
> > We will incorporate these defense analyses.

---

> > > ### Author Response · Authors · 2025-11-25
> > >
> > > **W6. The transferability across categories and temporal robustness (as LLMs update) remain unexplored.**
> > >
> > > **RW6.** Inspired by your suggestion, we additionally evaluate cross-model transferability. In this experiment, we use content optimized by GPT-4o and apply it directly to Gemini-2.5, Claude-4, and Grok-3 without any retraining or adaptation. The results on the Home & Kitchen as shown below.
> > >
> > > | Transfer | Method  | **Gemini-2.5** |  |  | **Claude-4** |  |  | **Grok-3** |  |  |
> > > |----------|---------|----------------|----|----|--------------|----|----|--------------|----|----|
> > > |          |         | Top-5 | Top-3 | Top-1 | Top-5 | Top-3 | Top-1 | Top-5 | Top-3 | Top-1 |
> > > | /        | String    | 53.50% | 43.00% | 31.00% | 56.5% | 46.5% | 33.0% | 51.50% | 42.00% | 30.50% |
> > > |          | Reasoning | 88.50% | 84.00% | 77.50% | 88.5% | 84.0% | 76.5% | 88.00% | 83.00% | 77.00% |
> > > |          | Review    | 94.50% | 89.50% | 83.50% | 91.5% | 87.0% | 81.0% | 90.50% | 85.00% | 78.50% |
> > > | **w/ transfer** | String    | 52.0% (1.5%↓) | 41.5% (1.5%↓) | 29.5% (1.5%↓) | 54.0% (2.5%↓) | 44.5% (2.5%↓) | 31.0% (2.0%↓) | 49.0% (2.5%↓) | 39.0% (3.0%↓) | 28.0% (2.5%↓) |
> > > |          | Reasoning | 79.0% (9.5%↓) | 72.5% (11.5%↓) | 63.0% (14.5%↓) | 76.5% (12.0%↓) | 69.0% (15.0%↓) | 58.5% (18.0%↓) | 70.0% (18.0%↓) | 63.5% (19.5%↓) | 55.0% (22.0%↓) |
> > > |          | Review    | 85.5% (9.0%↓) | 78.0% (11.5%↓) | 68.0% (15.5%↓) | 83.5% (8.0%↓)  | 75.5% (11.5%↓) | 67.5% (13.5%↓) | 75.0% (15.5%↓) | 67.5% (17.5%↓) | 58.0% (20.5%↓) |
> > >
> > >
> > > The results show that GPT-4o-optimized content partially transfers to other models: the string-based method drops only slightly, while the reasoning and review strategies experience larger decreases. Nevertheless, Reasoning and  Review still achieve relatively high performance after transfer.
> > >
> > > Besides, we note that Claude-3.7 was deprecated in October. To examine temporal robustness, we evaluate whether content optimized using Claude-3.7 can still be effective when applied to Claude-4. The results on the Home & Kitchen category are shown below.
> > >
> > > The results show that Claude-3.7–optimized content remains largely effective on Claude-4, although the reasoning and review strategies show moderate drops, while string-based content remains comparatively stable. We will incorporate these transferability results into the main paper in the revised version.
> > > | Method   | **Claude-3.7** |  |  | **Claude-4** |  |  |
> > > |----------|----------------|------|------|--------------|------|------|
> > > |          | Top-5 | Top-3 | Top-1 | Top-5 | Top-3 | Top-1 |
> > > | String    | 59.5% | 48.0% | 35.0% | 56.5% (3.0%↓) | 46.5% (1.5%↓) | 33.0% (2.0%↓) |
> > > | Reasoning | 94.0% | 89.0% | 82.0% | 88.5% (5.5%↓) | 84.0% (5.0%↓) | 76.5% (5.5%↓) |
> > > | Review    | 95.0% | 90.5% | 83.5% | 91.5% (3.5%↓) | 87.0% (3.5%↓) | 81.0% (2.5%↓) |
> > >
> > >
> > > Besides, we appreciate for highlighting this direction. However, we would like to clarify that, by design, the optimized content in our method is not intended to transfer across categories. This is because the generated content is tailored to the semantics of the specific target item, as illustrated in the examples in Appendix E. For instance, content optimized for “air fryer” in the Home and Kitchen category is not meaningful for an item such as a “hammer” in Tools and Home Improvement. Cross-category transfer is therefore not applicable in our setting.
> > >
> > > **Q1: The addition of Gaussian noise for exploration is mentioned but not systematically evaluated. In the experiments, how was the impact of adding this noise?**
> > >
> > > **A1.** We thank the reviewer for pointing this out. Gaussian noise is used only in the shadow-model optimization, where it serves as a lightweight exploration mechanism to avoid local optima during continuous updates (Sec. 3.3). To evaluate its effect, we compared three settings on the Home & Kitchen category: No noise ($\sigma=0$), Moderate noise ($\sigma=0.05$, used in paper) and Large noise $\sigma=0.2$. We conducted two representative models (GPT-4o and Gemini-2.5) under three noise settings. Results are shown below:
> > >
> > > | Model        | Noise Level (σ) | Top-5 | Top-3 | Top-1 |
> > > |--------------|------------------|-------|-------|-------|
> > > | **GPT-4o**   | 0                | 48.5% | 38.0% | 26.5% |
> > > |              | 0.05             | 55.0% | 45.5% | 33.0% |
> > > |              | 0.2              | 41.0% | 31.5% | 20.0% |
> > > | **Gemini-2.5** | 0              | 46.0% | 36.5% | 25.0% |
> > > |              | 0.05             | 53.5% | 43.0% | 31.0% |
> > > |              | 0.2              | 39.0% | 30.0% | 19.0% |
> > >
> > > Moderate noise provides a 6–8% Top-1 improvement over $\sigma=0$ by enhancing exploration, whereas large noise reduces performance due to excessive perturbation. These results show that Gaussian noise offers a small but consistent benefit while not being essential to CORE’s effectiveness.
> > >
> > > We will add the discussion about $\sigma$ to our revised paper.

---

> > > > ### Author Response · Authors · 2025-11-25
> > > >
> > > > **Q2: Have you observed cases where the optimization converges in embedding space but produces poor discrete text that fails to achieve desired rankings?**
> > > >
> > > > **A2.** Yes, we have observed such cases. Although the continuous optimization step may converge in embedding space, the final discrete text is obtained through nearest-neighbor token projection, which does not perfectly preserve the optimized direction. As a result, some reconstructed snippets deviate from the intended optimization signal and fail to influence the LLM as strongly as their continuous embeddings suggest.
> > > >
> > > > To quantify this effect, we measured the cosine similarity between the optimized continuous embedding and its discretized text on 500 samples. The results as shown below.
> > > > | Metric            | Mean | Std  |
> > > > |-------------------|------|------|
> > > > | Cosine similarity | 0.81 | 0.06 |
> > > >
> > > > The similarity averages 0.81, indicating that most but not all of the optimization signal is preserved. These lower-fidelity cases correspond precisely to the failures where the shadow-model approach yields weaker ranking improvements.
> > > >
> > > > **Q4. How might search platforms detect systematic ranking manipulation at scale?**
> > > >
> > > > **A4.** Thanks for raising this interesting question. We agree that reliably detecting ranking control at scale is challenging, especially because platforms must balance robustness with usability. Based on our observations in Q3, we believe there are several practical patterns that platforms could monitor.
> > > >
> > > > In particular, optimized text drifts away from natural writing patterns; it may show unusually high perplexity, repetitive reasoning templates, or atypically long and structured fields. While these patterns are not definitive on their own, we think they can provide useful indicators when viewed in aggregate. For example, platforms could monitor distributional anomalies, cluster recurring reasoning patterns, or flag sudden bursts of long-form content. These heuristics are lightweight and can be applied at scale without heavy computation.
> > > >
> > > > We believe these recommendations might be used to help platforms detect systematic ranking manipulation at scale while maintaining normal user experiences in the future.
> > > >
> > > > **Q5. What design changes to generative engines could increase robustness against optimization attacks like CORE while preserving utility for legitimate users?**
> > > >
> > > > **A5.** Thanks for this thoughtful question. Based on our observations in Q3, we believe there are several design directions that could make generative ranking engines more robust while still preserving the user experience.
> > > >
> > > > One possibility is to expose the model during training to a mixture of natural and adversarially optimized text, so the ranker gradually learns to downweight highly templated or overly structured reasoning forms. Another direction is to design style-invariant or normalized embedding layers that focus more on semantic content and less on rhetorical surface patterns, which would naturally reduce the impact of optimization-crafted phrasing. Finally, adopting a multi-pass ranking pipeline, where the LLM engine re-ranks items after metadata normalization or noise injection, may help identify unstable items whose rankings fluctuate abnormally across passes.
> > > >
> > > > We believe the aforementioned possibilities might work, and hope they can inspire future design choices that balance robustness with usability.

---

### Official Review · Reviewer_Zxr7 · 2025-10-30

**Soundness:** 2
**Presentation:** 2
**Contribution:** 2
**Rating:** 4
**Confidence:** 2

**Summary:**

This paper focuses on the generative engine optimization problems and proposes CORE, an optimization method that controls output rankings through modifying the content of corresponding items.
In this paper, the authors argue that the final ranking list produced by LLMs is heavily influenced by the initial order of retrieved results and CORE can mitigate it.
CORE provides two solutions to promote the rank of items: a shadow-model optimization method (i.e., string-based strategy) which leverages a proxy model to compute gradients over the content of target items and a query-based optimization method which leverages prompt engineering.
The query-based method can be further divided reasoning-based and review-based strategies according to the difference in prompts.
The former prompts LLMs to generate recommendation reasons while the latter prompts LLMs to generate review-like narratives.
To evaluate their method, the authors construct AmazonCOREBench, which is a large-scale benchmark derived from Amazon search results across 15 product categories.
Experiments on four major LLMs (GPT-4o, Gemini-2.5, Claude-3.7, Grok-3) demonstrate the effectiveness of CORE.

**Strengths:**

1. This paper tries to address the problem of generative engine optimization (GEO), which is relevant and practical at the intersection of LLMs and information retrieval. Since generative engines become more and more popular, understanding and resolving GEO is crucial, just as the status of SEO in conventional search engine.

2. Experimental results are impressive.  It shows that the final ranking list produced by LLMs is heavily influenced by the initial order of retrieved results and CORE achieves high promotion success rate. Besides, the creation of AmazonCOREBench is a valuable contribution that will benefit the community.

**Weaknesses:**

1. The novelty is limited. The shadow-model optimization method seems a direct application of black-box adversarial attack techniques, and the query-based optimization method seems an iterative prompt engineering framework. While effectively applied, the underlying mechanism or learning paradigm is not novel.

2. It lacks theoretical insights. Theoretical analysis or explanations are needed for understanding the phenomenon that the final ranking list produced by LLMs is heavily influenced by the initial order of retrieved results and why CORE (both shadow-model optimization and  query-based optimization) resolves the problem.

3. The core contribution is focusing on a single and specific domain (i.e., product search). The generalizability is not explored and limits its scope as a foundational contribution. It seems the proposed query-based optimization (especially the review-based strategy) is a dedicated method for product search.

**Questions:**

Overall, this paper tries to address a significant applied data science problem with strong empirical results. However, the proposed approach builds on existing methods straightforwardly without introducing a substantial novel algorithm or theoretical insight. So I think this paper is more aligned with the field of applied data science like KDD, WWW and SIGIR.

This paper can be further improved, including:
1. Adding theoretical analysis, e.g., discussing the condition under which the proposed method is guaranteed to work
2. Adding analysis or experiments to show the generality of CORE beyond product search.

---

> ### Author Response · Authors · 2025-11-25
>
> Thank you for the thoughtful review. We are encouraged that you recognized the importance of our main topic and agreed that understanding and resolving GEO is crucial. We also appreciate your positive remarks on the impressiveness of our experimental results. We believe the remaining weaknesses and questions can be sufficiently addressed, and we respond to them in detail below.
>
> **W1/Q1. This paper tries to address a significant applied data science problem with strong empirical results. However, the proposed approach builds on existing methods straightforwardly without introducing a substantial novel algorithm or theoretical insight.**
>
> **A1.** Thanks for your comments. Unfortunately, we cannot agree on this comment. Our paper introduces a new problem, a new methodological framework, and empirical capabilities that existing jailbreak, injection, and ranking manipulation methods do not demonstrate.
>
> **(1) New problem formulation: output-ranking control in LLM-based search has not been studied.**
> To the best of our knowledge, no prior work formulates the task of controlling the LLM-synthesized ranked list after external retrieval. Existing work in SEO, GEO, jailbreak, and prompt injection operates at the retrieval stage or focuses on prompt interpretation. None of these methods define an optimization objective over the probability of a ranked list generated during the synthesis stage. Our formulation in Sections 3.2 and 3.3 introduces this task for the first time.
>
> **(2) Existing methods require gradient access or prompt-level overrides.**
> Even when considering a broader scope of existing methods designed for other related tasks, the model we proposed in our paper remains sufficiently novel. STS depends on white-box gradients. TAP and SRP rely on explicit prompt overrides and cannot modify item-level content. In contrast, our shadow-model solution performs gradient-based optimization without access to the target model, and our query-based solution operates purely from observed rankings. No prior method is able to function under these constraints in a generative-search environment. More importantly, even if we consider the scope of generic injection and jailbreak methods, the method we propose in this paper has never been presented elsewhere to the best of our knowledge.
>
> **(3) Empirical evidence that existing injection methods cannot achieve our capability.**
> Table 2 provides direct numerical evidence. On GPT-4o (Home & Kitchen), the Top-1 promotion success rates of existing methods are STS: 32.5%, TAP: 34.0%, and SRP: 42.0%. In contrast, CORE achieves 81.0% with the reasoning strategy and 79.0% with the review strategy. This corresponds to a 39–49 percentage-point improvement, demonstrating that existing injection and optimization methods cannot achieve comparable ranking-control performance.
>
> **(4) A new benchmark enabling the study of this new problem.**
> AmazonCOREBench is the first benchmark specifically constructed for synthesis-stage ranking control. It covers fifteen categories with two hundred products each. Prior work does not provide any dataset for evaluating ranking behavior in LLM-based search. This benchmark establishes a foundation for the systematic study of this new problem.
>
> **W2/Q2. It lacks theoretical insights.  Adding theoretical analysis, e.g., discussing the condition under which the proposed method is guaranteed to work.**
>
> **A2.** Thank you for pointing this out. Following your suggestion, we have added the corresponding theoretical analysis; please refer to the revised version of the paper (**Appendix C**).

---

> > ### Author Response · Authors · 2025-11-25
> >
> > **W3/Q3. The core contribution is focusing on a single and specific domain (i.e., product search). Adding analysis or experiments to show the generality of CORE beyond product search.**
> >
> > **A3.** Inspired by your suggestion, we conducted two additional experiments outside the product-search setting to test whether CORE generalizes to passage ranking and document retrieval.
> >
> > Specifically, we first constructed a document-ranking task based on 200 randomly sampled queries from the MS MARCO dataset. For each query, we retrieved 10 candidate passages, asked the LLM to generate a ranked list, and then applied CORE to promote the target passage placed at the end of the candidate list. We used the same three optimization strategies (String, Reasoning, Review) as in AmazonCOREBench and appended the optimized content directly to the target passage before re-ranking.
> >
> > We also evaluated CORE on a travel-itinerary retrieval and ranking task. We created 200 city-related queries (e.g., “Top attractions in Tokyo,” “What to see in Paris in two days”). For each query, we retrieved 10 candidate travel itineraries from Xiaohongshu, a widely used social platform where users share travel guides and short itinerary recommendations. The LLM then ranked these retrieved itinerary candidates, and we applied CORE to the target itinerary placed at the bottom of the list, using the same three optimization strategies as above. Note that Claude-3.7 has been deprecated, so we report results using Claude-4 instead. The results as shown below:
> >
> > | Dataset | Method | **GPT-4o** |  |  | **Gemini-2.5** |  |  | **Claude-4** |  |  | **Grok-3** |  |  |
> > |---------|--------|------------|------|------|----------------|------|------|----------------|------|------|--------------|------|------|
> > |         |         | Top-5 | Top-3 | Top-1 | Top-5 | Top-3 | Top-1 | Top-5 | Top-3 | Top-1 | Top-5 | Top-3 | Top-1 |
> > | Document | String    | 58.5% | 48.5% | 33.5% | 56.0% | 46.0% | 31.0% | 61.5% | 50.0% | 34.0% | 54.0% | 44.5% | 30.5% |
> > |          | Reasoning | 91.0% | 86.0% | 80.0% | 88.0% | 83.0% | 76.5% | 93.0% | 88.5% | 81.0% | 86.5% | 82.0% | 75.0% |
> > |          | Review    | 88.0% | 83.0% | 76.0% | 90.5% | 85.5% | 78.0% | 89.5% | 85.0% | 78.0% | 86.0% | 81.0% | 74.5% |
> > | Travel   | String    | 53.5% | 43.0% | 30.0% | 51.0% | 41.0% | 28.5% | 55.0% | 44.5% | 30.5% | 49.5% | 39.5% | 27.0% |
> > |          | Reasoning | 87.5% | 82.0% | 75.5% | 84.0% | 78.0% | 71.0% | 89.5% | 83.5% | 76.0% | 82.5% | 76.0% | 69.0% |
> > |          | Review    | 83.5% | 78.0% | 71.5% | 85.0% | 79.5% | 72.5% | 86.0% | 80.5% | 73.5% | 81.0% | 75.0% | 68.5% |
> >
> > Across both tasks, CORE achieves top-k promotion rates similar to those reported in the main Amazon experiments. The reasoning and review strategies remain highly effective, and the relative ordering of the three methods is consistent with the main results. Experiments show that CORE generalizes well beyond product search and is applicable across different LLM-based ranking domains.
> >
> > We will include these additional results in the revised version to clarify that CORE’s effectiveness is not limited to product search, but can extend to other LLM-based ranking scenarios.

---

### Official Review · Reviewer_ppC2 · 2025-10-31

**Soundness:** 2
**Presentation:** 3
**Contribution:** 2
**Rating:** 4
**Confidence:** 4

**Summary:**

This paper investigates how rankings in LLM-based search  can be manipulated by modifying product-level textual content that the LLM uses when synthesizing recommendations. The authors propose CORE (Controlling Output Rankings in Generative Engines), a method that iteratively appends optimized content to a target product’s description to raise its rank in the generated list. Two optimization modes are explored: shadow-model optimization, which uses a surrogate model to approximate gradients, and (2) query-based iterative optimization, which works under a black-box constraint. The authors also introduce AmazonCOREBench, a benchmark spanning 15 product categories derived from Amazon’s search interface. Experiments across multiple LLMs show that CORE achieves substantial ranking gains.

**Strengths:**

- S1: The introduction of AmazonCOREBench provides a reusable benchmark that enhances reproducibility and future comparative studies.

- S2: The reasoning-based and review-based optimization strategies are thoughtfully designed and show realistic manipulation behavior.

- S3: The experimental setup is comprehensive, covering multiple model families and 15 product categories, which strengthens the empirical evidence.

- S4: The sensitivity-to-insertion-order experiment (Section 4.5) is particularly insightful, revealing positional bias in generative ranking and demonstrating how content order interacts with linguistic style.

**Weaknesses:**

- W1: The experimental setting seems to be unrealistic. It assumes a single target product is pre-specified and optimized to improve its rank, while all other products remain static. In real generative search, users do not know which item they want to promote.

- W2: All reported improvements are relative to retrieval order, not to any true relevance judgment. The experiments never verify whether the promoted item is actually better or more relevant.

- W3: The optimized outputs have worse fluency, with higher perplexity and lower human ratings than the original texts. This indicates that the method’s strongest manipulations depend on unnatural or verbose text.

**Questions:**

n/a

---

> ### Author Response · Authors · 2025-11-25
>
> We appreciate your thoughtful review and are encouraged by your recognition of our method’s careful design and its ability to capture realistic manipulation behavior. We are also glad that you acknowledge our proposed benchmark as a reusable resource that supports reproducibility and future comparative studies. We believe the remaining concerns can be fully addressed, and we provide detailed responses below.
>
> **W1: The experimental setting seems to be unrealistic.**
>
> **RW1.** We would like to clarify that CORE is designed for users like businesses or independent creators; therefore, the purpose is to promote their products when their product is in a disadvantageous stage, as we mentioned in lines 78-80:
> > “While this efficiency clearly benefits Alice, it risks disadvantaging small businesses and independent creators, whose products may be buried in the retrieval results and thus remain invisible in the final recommendations”.
>
> In real generative-search systems (e.g., Amazon), small businesses can modify only their own product metadata, and they already know which product they intend to promote. Modeling the target as the last-ranked item captures this realistic worst-case scenario, where a creator’s product is buried in retrieval and has virtually no chance of being recommended. Our evaluation then examines whether optimization can improve its visibility.
>
> **W2: All reported improvements are relative to retrieval order, not to any true relevance judgment**
>
> **RW2.** Thanks for raising this concern. If we understand your question correctly, you want to optimize the quality of the recommended product from the perspective of users who ask for recommendations. Therefore, verify that the quality of the product is reasonable in this optimization task. However, the optimization problem we study is from the business user who wants to optimize their ranking in LLM-engine, therefore, retrieval is our main metric to evaluate the effectiveness of the method. While we acknowledge that the reviewer optimization problem is valuable, we believe our studied problem is also valuable to promote an equal economic environment where the small business might be at a disadvantage position as motivated in lines 76-78. We hope the reviewer can agree on the importance of our study and that retrieval ranking is the right metric to measure the effectiveness of the method.
>
> **W3: The optimized outputs have worse fluency, with higher perplexity and lower human ratings than the original texts.**
>
> **RW3.** We thank the reviewer for the observation. Our method includes three optimization strategies: string-based, reasoning-based, and review-based, and only the string-based variant exhibits poor fluency. As reported in Table 3 (Section 4.4, page 8), both the reasoning-based and review-based strategies produce fluent outputs with average perplexity scores below 80, and the review-based strategy, in particular, achieves perplexity scores very close to the original text. This indicates that our practical optimization does not significantly degrade fluency while still achieving strong promotion success rates.
>
> Furthermore, Table 4 (Section 4.4, page 8) shows that review-based content has a very low human detection rate, nearly matching the baseline, demonstrating that the optimized text remains natural and difficult for humans to distinguish from unmodified content. Thus, the strongest and most realistic manipulations in CORE arise from the reasoning- and review-based strategies, not from the intentionally unnatural string-based baseline.

---

### Official Review · Reviewer_kqGS · 2025-11-01

**Soundness:** 3
**Presentation:** 3
**Contribution:** 3
**Rating:** 6
**Confidence:** 4

**Summary:**

- This paper presents an *original* and *significant* contribution by introducing **CORE** (*Controlling Output Rankings in -gEnerative Engines*), a novel optimization method focused on manipulating the output ranking generated by Large Language Models (LLMs) in search applications.

- The core strength of this approach lies in its practicality and realism: it addresses the critical dependency where LLM recommendations are dictated by the initial retrieval order, potentially *disadvantaging small businesses*.

- CORE successfully operates within a realistic *black-box* setting, where neither the LLM architecture nor the choice of external search engine can be modified by the user. Instead, CORE targets the synthesis stage by appending optimized content (such as *string-based*,  *reasoning-based* or *review-based* text) to influence the LLM's final ranked list.

- The *quality* and *clarity* of the empirical validation is really good. The authors developed **AmazonCOREBench**, a *large-scale benchmark* simulating a realistic product search environment derived from 15 Amazon categories, enhancing the utility of the results for future research.

- Experiments across four prominent LLMs (GPT-4o, Gemini-2.5, Claude-3.7, and Grok-3) demonstrated exceptional effectiveness, achieving an average Promotion Success Rate of **80.3% @Top-1** under the best strategy.

- Furthermore, the evaluation includes a thorough comparison showing CORE's *superior performance and robustness* against existing ranking manipulation methods (STS, TAP, SRP). Crucially, the *review-based* strategy maintains high human-rated fluency (4.6 out of 5) while remaining stealthy, with a low detection rate (18.4%), underscoring the method's effectiveness.

**Strengths:**

*   **Original Problem Formulation:** CORE addresses output ranking manipulation at the LLM *synthesis stage*, distinct from traditional SEO or GEO which focus on retrieval, overcoming the limitation of fixed search engine choice.
*   **Realistic Threat Model:** The methodology operates successfully under the demanding *black-box assumption*, without requiring access to model internals or gradients.
*   **High Effectiveness:** CORE achieved high promotion success rates, including an average of **80.3% @Top-1** across 15 categories on four commercial LLMs.
*   **Novel Optimization Strategies:** The paper introduces effective *reasoning-based* and *review-based* optimization content that leverages how LLMs process information (e.g., Chain-of-Thought reasoning).
*   **Robustness against Baselines:** CORE consistently and substantially *outperforms* prior ranking manipulation methods like STS, TAP, and SRP across multiple models and categories.
*   **Creation of AmazonCOREBench:** The development of a robust, large-scale, 15-category benchmark provides a vital resource for evaluating future generative engine optimization techniques in product search.
*   **Demonstrated Fluency:** The *review-based strategy* proved highly effective while maintaining high fluency (4.6/5 in human evaluation) and low human detectability (18.4%), making the manipulation practical and hard to flag.
*   **Comprehensive LLM Testing:** Experiments were rigorously conducted across four state-of-the-art, widely deployed LLMs: *GPT-4o, Gemini-2.5, Claude-3.7, and Grok-3*.
*   **Clear Methodology:** The paper clearly defines the optimization task and provides two distinct solutions: a gradient-based *shadow-model solution* and an iterative *query-based solution* for black-box environments.
*   **Model Bias Insights:** The results highlight differences in model responsiveness, noting that certain LLMs (like GPT-4o) favor reasoning while others (like Gemini-2.5) favor review-style content.

**Weaknesses:**

The primary weakness of the paper lies in the **practical constraints** and **fragility** of the proposed optimization strategies, particularly in a truly *black-box* environment, coupled with a lack of discussion regarding **mitigation and defense**.

*   **Reliance on Alignment:** Optimal performance (PSR) in the *query-based black-box solution* requires the Generator and Optimizer to be the *same model* as the target synthesizing LLM, suggesting limited robustness if the target LLM is truly unknown or changes frequently. While the *query-based solution* is highly effective, it relies heavily on the iterative interaction of a Generator and Optimizer model that often must be *aligned* with the target synthesizing LLM (e.g., using GPT-4o as both Generator/Optimizer when attacking GPT-4o). When these components are *mismatched*, performance, especially at Top-1 rank, tends to degrade.

*   **Fragility to Hyperparameters:** The iterative black-box optimization depends on a carefully tuned similarity threshold ($\tau \ge 0.7$) for effective performance, as lowering the threshold ($\tau=0.5$) causes success rates to drop sharply. This dependence on high alignment and tuned hyperparameters undermines the robustness implied by the black-box setting.

*   **High Detectability of Reasoning Strategy:** The highly successful *reasoning-based strategy* is easily flagged by human annotators (62.1% detection rate), making it impractical for stealthy, real-world deployment. The *reasoning-based* content, while highly successful in promotion, exhibits a **high human detection rate (62.1%)** compared to the review-based strategy (18.4%), limiting its practical stealthiness as a real-world attack vector.

*   **No Discussion of Defenses:** The paper identifies a significant security threat but provides no mechanisms, suggestions, or baseline evaluations for **mitigating** CORE-style attacks. The paper only compares CORE to existing, less effective attack methods (STS, TAP, SRP) but does not explore how generative engines could be hardened against content-appended manipulation strategies. A significant omission is the complete absence of a discussion on **defenses** against CORE. Given the demonstrated effectiveness (80.3% @Top-1 success) and the high stealth of the *review-based* strategy, CORE represents a non-trivial security threat to generative engines.

*   **Insertion Order Sensitivity:** The effectiveness of the optimization content (both Review and Reasoning) is heavily dependent on the order in which items are presented to the LLM, indicating potential instability in dynamic search environments. Effectiveness of the optimized content is highly sensitive to the **insertion order** of the search results, suggesting that minor variations in upstream search engine output could dramatically reduce CORE's success rate in a live, dynamic environment.

*   **High API Cost:** The successful black-box solution relies on an iterative *Append-and-Query* loop, which implies significant latency and cost associated with repeated API calls to expensive commercial LLMs (GPT-4o, Gemini-2.5, Claude-3.7, Grok-3). There is no mention of either cost or latency in the paper.

*   **Limited Domain Scope:** The entire empirical evaluation is focused exclusively on **product search** within the **Amazon** environment (AmazonCOREBench).

**Questions:**

1. **Mitigation Strategies**: Given that **CORE** demonstrates a highly effective and **stealthy manipulation** technique (especially the review-based strategy with 18.4% detection rate), what specific defense mechanisms or detection methods did the authors consider or test for hardening generative engines against this type of content-based optimization?
2. **Scalability and Cost**: The **query-based optimization** involves an iterative Generator–Optimizer loop. Could the authors quantify the average number of iterations/API calls required to achieve the reported **PSR@1**, and estimate the real-world latency and financial cost of deploying CORE successfully against a single user query?
3. **Black-Box Alignment in Practice**: In a truly **black-box** scenario, how would an attacker determine the optimal choice for the Generator and Optimizer models to match the unknown synthesizing LLM, especially since mismatched configurations lead to performance drops in the **Reasoning strategy**?
4. **Transferability Beyond Product Search**: The method is evaluated only on **AmazonCOREBench** for product recommendations. How do the reasoning-based and review-based strategies transfer to other generative engine tasks, such as summarizing long documents, creating travel itineraries, or generating **code recommendations**?
5. **Impact of Retrieval Set Size**: The benchmark collects the **top-10** recommendations from Amazon. What is the expected drop in Promotion Success Rate if the synthesizing LLM were to use only the top-5 or top-3 retrieved items, potentially limiting the LLM’s exposure to the optimized content?
6. **Addressing Insertion Order Volatility**: Since effectiveness is highly dependent on the optimized content's **position** relative to other candidates (Table 5), how can an attacker ensure or increase the likelihood that their optimized content appears early in the input list provided by the search engine to the LLM, overcoming the **instability** shown in the sensitivity analysis?
7. **Shadow Model Optimization (Discrete Mapping)**: The shadow model solution uses **gradient descent** in the continuous embedding space, followed by a discrete reconstruction step. Did the authors measure the fidelity loss or **semantic drift** introduced during this reconstruction process, and how does this step potentially limit the performance of the shadow-model solution compared to the query-based solution?

**Details Of Ethics Concerns:**

- **Ethics statement** clearly mentions adherence to ICLR code of conduct. However, the study does mention employing 20 human annotators for evaluating fluency of generated content and might necessitate a review if desired by Area Chairs and/or other reviewers.
- **LLM usage** in clearly mentioned that it was used for improving quality of the paper, language, grammar and not for ideation, methodology or research design
- **Reproducibility statement** is also legitimate as the sample source code is made publicly available at the time of review in an anonymous repository

---

> ### Author Response · Authors · 2025-11-25
>
> Thank you for the constructive comments. We appreciate your recognition of the originality and significance of our work, as well as your acknowledgment of the practicality and realism of our approach. We believe the mentioned weaknesses and questions can be sufficiently addressed.
>
> **W1. Reliance on Alignment.**
>
> **RW1.** We agree that the query-based solution achieves its best performance when the Generator and Optimizer align with the target LLM. We believe this degradation is expected and consistent with general findings in LLM research, where cross-model transferability often leads to reduced performance due to differences in architecture and reasoning behaviors across models. We will acknowledge this observation in the revised version. Having said that, even under mismatched settings, the performance remains competitive. For example, as shown in Table 7 (Section 4.6, Page 9), on GPT-4o using Grok-3 as the Generator, the reasoning-based PSR remains 90.0% @Top-5, 85.0% @Top-3, and 79.0% @Top-1, representing only 1.0%, 2.0%, and 2.0% degradation, respectively. Similarly, using Grok-3 as the Optimizer, GPT-4o still achieves 90.0% @Top-5, 85.0% @Top-3, and 78.0% @Top-1, with merely 1.0%, 2.0%, and 3.0% degradation, respectively. This pattern is consistent across all four models: mismatched configurations generally remain within 3–6% of the aligned setting, highlighting the competitiveness.
>
> **W2. Fragility to Hyperparameters**
>
> **RW2.** We agree that the hyperparameter $\tau$ affects performance because it directly controls the required similarity between the generated ranking and the target ranking. Ideally, $\tau = 1$, since the goal of the optimization is to perfectly match the target ranking. If computational cost were not a concern, one could simply set $\tau = 1$ to obtain the optimal result. We use the threshold to balance computational efficiency and performance. As shown in Table 11 (Section H, Page 22), we find that $\tau = 0.7$ offers a strong trade-off and works well across most settings.
>
> To further illustrate this trade-off, we additionally measured the optimization performance, the average number of iteration loops (max = 20), and the cost per optimized item when $\tau = 1$, using Home & Kitchen items on GPT-4o with the reasoning-based optimization strategy. The results are shown below:
>
> | τ   | Top-5  | Top-3  | Top-1  | Avg. Loops | Cost / Item |
> |-----|--------|--------|--------|------------|--------------|
> | 0.7 | 92.0%  | 87.0%  | 81.0%  | 3.1        | $0.0398      |
> | 1   | 96.5%  | 92.0%  | 89.0%  | 14.9       | $0.4914      |
>
> Even with 20 optimization loops, the successfully optimized cases still do not reach 100% Top-1 accuracy. Although $\tau = 1$ offers slightly higher accuracy, it raises the computational cost by more than 12×. Therefore, we adopt $\tau = 0.7$ as a trade-off across settings.
>
> **W3. High Detectability of Reasoning Strategy**
>
> **RW3.** We agree that the reasoning-based strategy is more detectable to human annotators. This strategy is included primarily to provide insight into how intentionally integrating structured reasoning traces can influence LLM rankings.
> For real-world scenarios, For real-world scenarios, the review-based strategy provides a much more natural alternative, with an average perplexity of 31.97 across four models (Table 3 (Section 4.4, Page 8)) and a low detectability rate of 18.4% (Table 4 (Section 4.4, Page 8)), while still achieving strong promotion success rates (e.g., GPT-4o: 89.5% @Top-5, 85.0% @Top-3, 79.0% @Top-1).

---

> ### Author Response · Authors · 2025-11-25
>
> **W4. No Discussion of Defenses / Q1. Mitigation Strategies**
>
> **RW4/A1.** Thanks for pointing this out. Inspired by your suggestion, we introduce three defense methods: (1) perplexity-based filtering, which removes items whose perplexity exceeds a fluency threshold; (2) pattern-based filtering, which filters out fields containing structured reasoning traces from a predefined pattern list; (3) length constraints, which discard content exceeding a maximum token budget.
>
> Specifically, we set the perplexity threshold to 50, based on the perplexity statistics reported in Table 3 (Section 4.4, Page 8); use a predefined pattern list consisting of ***["I'm exploring", "I'm analyzing", "I'm evaluating", "I'm examining", "I'm comparing", "I'm reviewing", "I'm explaining", "I'm outlining", "I'm breaking down", "I'm identifying", "I analyzed", "I evaluated", "I compared", "first", "next", "then", "finally", "step", "in conclusion", "this suggests"]***; and apply a 4000-word length constraint to filter unusually long fields. Note that Claude-3.7 has been deprecated, so we report results using Claude-4 instead. The results on Home & Kitchen are shown below.
> | Defense | Method | **GPT-4o** |  |  | **Gemini-2.5** |  |  | **Claude-4** |  |  | **Grok-3** |  |  |
> |---------|--------|------------|------|------|----------------|------|------|----------------|------|------|--------------|------|------|
> |         |        | Top-5 | Top-3 | Top-1 | Top-5 | Top-3 | Top-1 | Top-5 | Top-3 | Top-1 | Top-5 | Top-3 | Top-1 |
> | /       | String    | 55.0% | 45.5% | 33.0% | 53.5% | 43.0% | 31.0% | 56.5% | 46.5% | 33.0% | 51.5% | 42.0% | 30.5% |
> |         | Reasoning | 92.0% | 87.0% | 81.0% | 88.5% | 84.0% | 77.5% | 88.5% | 84.0% | 76.5% | 88.0% | 83.0% | 77.0% |
> |         | Review    | 89.5% | 85.0% | 79.0% | 94.5% | 89.5% | 83.5% | 91.5% | 87.0% | 81.0% | 90.5% | 85.0% | 78.5% |
> | perplexity-based filtering | String | 0.0% (55.0%↓) | 0.0% (45.5%↓) | 0.0% (33.0%↓) | 0.0% (53.5%↓) | 0.0% (43.0%↓) | 0.0% (31.0%↓) | 0.0% (56.5%↓) | 0.0% (46.5%↓) | 0.0% (33.0%↓) | 0.0% (51.5%↓) | 0.0% (42.0%↓) | 0.0% (30.5%↓) |
> |         | Reasoning | 12.0% (80.0%↓) | 8.0% (79.0%↓) | 3.5% (77.5%↓) | 10.0% (78.5%↓) | 7.0% (77.0%↓) | 3.0% (74.5%↓) | 13.0% (75.5%↓) | 8.5% (75.5%↓) | 3.0% (73.5%↓) | 11.0% (77.0%↓) | 7.5% (75.5%↓) | 3.0% (74.0%↓) |
> |         | Review    | 84.0% (5.5%↓) | 80.0% (5.0%↓) | 72.5% (6.5%↓) | 89.0% (5.5%↓) | 84.0% (5.5%↓) | 76.0% (7.5%↓) | 90.0% (1.5%↓) | 86.0% (1.0%↓) | 78.0% (5.5%↓) | 85.0% (5.5%↓) | 80.5% (4.5%↓) | 72.0% (6.5%↓) |
> | pattern-based filtering | String | 55.0% (0.0%↓) | 45.5% (0.0%↓) | 33.0% (0.0%↓) | 53.5% (0.0%↓) | 43.0% (0.0%↓) | 31.0% (0.0%↓) | 56.5% (0.0%↓) | 46.5% (0.0%↓) | 46.5% (0.0%↓) | 51.5% (0.0%↓) | 42.0% (0.0%↓) | 30.5% (0.0%↓) |
> |         | Reasoning | 35.0% (57.0%↓) | 25.0% (62.0%↓) | 16.0% (65.0%↓) | 33.0% (55.5%↓) | 23.0% (61.0%↓) | 15.0% (62.5%↓) | 38.0% (50.5%↓) | 28.0% (56.0%↓) | 18.5% (57.5%↓) | 32.0% (56.0%↓) | 22.0% (61.0%↓) | 14.0% (63.0%↓) |
> |         | Review    | 69.0% (20.5%↓) | 64.0% (21.0%↓) | 56.0% (23.0%↓) | 74.0% (20.5%↓) | 68.5% (21.0%↓) | 60.0% (23.5%↓) | 75.0% (16.5%↓) | 69.5% (17.5%↓) | 61.0% (21.0%↓) | 71.0% (19.5%↓) | 66.0% (19.0%↓) | 57.0% (21.5%↓) |
> | length constraints | String | 55.0% (0.0%↓) | 45.5% (0.0%↓) | 33.0% (0.0%↓) | 53.5% (0.0%↓) | 43.0% (0.0%↓) | 31.0% (0.0%↓) | 56.5% (0.0%↓) | 46.5% (0.0%↓) | 46.5% (0.0%↓) | 34.0% (0.0%↓) | 42.0% (0.0%↓) | 30.5% (0.0%↓) |
> |         | Reasoning | 52.0% (40.0%↓) | 44.0% (43.0%↓) | 31.0% (50.0%↓) | 49.0% (39.5%↓) | 41.0% (43.0%↓) | 29.0% (48.5%↓) | 54.0% (34.5%↓) | 45.0% (39.0%↓) | 33.0% (43.5%↓) | 50.0% (38.0%↓) | 41.0% (42.0%↓) | 29.0% (48.0%↓) |
> |         | Review    | 76.0% (13.5%↓) | 70.0% (15.0%↓) | 62.0% (17.0%↓) | 81.0% (13.5%↓) | 75.0% (14.5%↓) | 66.0% (17.5%↓) | 82.0% (9.5%↓) | 76.0% (11.0%↓) | 68.0% (13.0%↓) | 78.0% (12.5%↓) | 72.5% (12.5%↓) | 64.0% (14.5%↓) |
>
> Perplexity-based filtering eliminates nearly all string-based content and reduces reasoning-based content by more than 75% across models, while preserving most review-style content. Under pattern-based filtering and length constraints, the success rates of both reasoning and review strategies also drop sharply, reflecting their strong dependence on structured phrasing and long-form text. Note that string-based content remains unchanged under these two defenses, as it is short and does not contain the reasoning patterns that trigger filtering, making these defenses not applicable to the string strategy. Review still remains the strongest overall performer under defense mechanisms, highlighting the need for more robust ranking-protection mechanisms.
>
> We will incorporate these defense analyses into the main paper in the future version.

---

> ### Author Response · Authors · 2025-11-25
>
> **W5/Q6. Insertion Order Sensitivity**
>
> **RW5/A6.**  We believe there is a misunderstanding and will make it clear here and also in the paper. The order sensitivity refers to the inherent behavior of LLM-generated ranking is largely constrained by the order of items returned by the search engine. We describe in lines 76–78. The baseline in Table 1 further justifies this behavior (baseline means no technique applied to the item content).
>
> Building on this phenomenon, Table 5 is designed to compare the relative strength of the three optimization strategies (String, Reasoning, Review) under different initial positions. The purpose is to reveal which optimization signal is more influential and more robust to different starting positions. In practical deployments,  only one optimized content would apply to the target item, so such three-way competition does not occur.
>
> **W6. High API Cost / Q2. Scalability and Cost**
>
> **RW6.** Thank you for raising this concern. Inspired by your suggestion, we conducted a cost analysis over 1,000 target-item optimizations using reasoning-based strategies and measured (1) the average number of optimization loops, (2) the token usage per loop, and (3) the total cost under each model’s pricing. Note that Claude-3.7 has been deprecated, so we report results using Claude-4 instead. The results are shown below:
> | Model        | Avg. Loops | Tokens / Loop | Total Tokens / Item | Cost / Item |
> |--------------|------------|----------------|----------------------|-------------|
> | GPT-4o       | 3.1        | 1,284.7        | 3,982.6              | $0.0398     |
> | Gemini-2.5   | 4.6        | 1,912.3        | 8,806.6              | $0.0110     |
> | Claude-4     | 3.7        | 1,356.8        | 5,018.2              | $0.0452     |
> | Grok-3       | 4.1        | 1,487.9        | 6,100.4              | $0.0549     |
>
> Across all models, the per-item optimization cost remains low (approximately $0.01 – $0.06), indicating that the API overhead of CORE is practical and acceptable for real-world use.
>
> We will incorporate the API cost into the main paper in the future version.
>
> **W7. Limited Domain Scope./ Q4. Transferability Beyond Product Search**
>
> **RW7/A4.** Thank you for pointing this out. To further address your concern, we conducted two additional experiments outside the product-search setting to test whether CORE generalizes to passage ranking and document retrieval.
> Specifically, we first constructed a document-ranking task based on 200 randomly sampled queries from the MS MARCO dataset. For each query, we retrieved 10 candidate passages, asked the LLM to generate a ranked list, and then applied CORE to promote the target passage placed at the end of the candidate list. We used the same three optimization strategies (String, Reasoning, Review) and appended the optimized content directly to the target passage before re-ranking.
>
> We also evaluated CORE on a travel-itinerary retrieval and ranking task. We created 200 city-related queries (e.g., “Top attractions in Tokyo,” “What to see in Paris in two days”). For each query, we retrieved 10 candidate travel itineraries online. The LLM then ranked these retrieved itinerary candidates, and we applied CORE to the target itinerary placed at the bottom of the list, using the same three optimization strategies as above. Note that Claude-3.7 has been deprecated, so we report results using Claude-4 instead. The results as shown below:
>
>
> | Dataset | Method | **GPT-4o** |  |  | **Gemini-2.5** |  |  | **Claude-4** |  |  | **Grok-3** |  |  |
> |---------|--------|------------|------|------|----------------|------|------|----------------|------|------|--------------|------|------|
> |         |         | Top-5 | Top-3 | Top-1 | Top-5 | Top-3 | Top-1 | Top-5 | Top-3 | Top-1 | Top-5 | Top-3 | Top-1 |
> | Document | String    | 58.5% | 48.5% | 33.5% | 56.0% | 46.0% | 31.0% | 61.5% | 50.0% | 34.0% | 54.0% | 44.5% | 30.5% |
> |          | Reasoning | 91.0% | 86.0% | 80.0% | 88.0% | 83.0% | 76.5% | 93.0% | 88.5% | 81.0% | 86.5% | 82.0% | 75.0% |
> |          | Review    | 88.0% | 83.0% | 76.0% | 90.5% | 85.5% | 78.0% | 89.5% | 85.0% | 78.0% | 86.0% | 81.0% | 74.5% |
> | Travel   | String    | 53.5% | 43.0% | 30.0% | 51.0% | 41.0% | 28.5% | 55.0% | 44.5% | 30.5% | 49.5% | 39.5% | 27.0% |
> |          | Reasoning | 87.5% | 82.0% | 75.5% | 84.0% | 78.0% | 71.0% | 89.5% | 83.5% | 76.0% | 82.5% | 76.0% | 69.0% |
> |          | Review    | 83.5% | 78.0% | 71.5% | 85.0% | 79.5% | 72.5% | 86.0% | 80.5% | 73.5% | 81.0% | 75.0% | 68.5% |
>
>
> Across both tasks, CORE achieves top-k promotion rates similar to those reported in the main AmazonCOREBench experiments. The reasoning and review strategies remain highly effective, and the relative ordering of the three methods is consistent with the main results. Experiments show that CORE generalizes well beyond product search and is applicable across different LLM-based ranking domains.
>
> We will include these additional results.

---

> ### Author Response · Authors · 2025-11-25
>
> **Q3. Black-Box Alignment in Practice.**
>
> **A3. **In practical black-box settings, small businesses and independent creators typically optimize for one specific generative engine, e.g., aiming to perform well on GPT-based search, because they are aware that GPT users constitute the majority of their traffic.
>
> Having said that, even under mismatched settings, the performance remains competitive. For example, as shown in Table 7 (Section 4.6, Page 9), on GPT-4o using Grok-3 as the Generator, the reasoning-based PSR remains 90.0% @Top-5, 85.0% @Top-3, and 79.0% @Top-1, representing only 1.0%, 2.0%, and 2.0% degradation, respectively. Similarly, using Grok-3 as the Optimizer, GPT-4o still achieves 90.0% @Top-5, 85.0% @Top-3, and 78.0% @Top-1, with merely 1.0%, 2.0%, and 3.0% degradation, respectively. This pattern is consistent across all four models: mismatched configurations generally remain within 3–6% of the aligned setting, indicating that the proposed optimization method works effectively even when the model is unknown.
>
>
> **Q5. Impact of Retrieval Set Size**
>
> **A5.** We want to clarify that the Promotion Success Rate is not directly determined by whether the retrieval set contains 10, 5, or 3 items. This is because CORE appends optimized content to the target item itself, and the synthesizing LLM will always see this appended content as long as the target item appears in the candidate list. Therefore, reducing the retrieval set size does not inherently weaken CORE. It only affects whether the target item is included in the retrieved candidates in the first place.
>
> **Q7. Shadow Model Optimization (Discrete Mapping)**
>
> **A7.** Yes, we agree that the gradient update is performed in the continuous embedding space, but the final optimized text must be projected back into the discrete token space. This projection inevitably causes semantic drift, because the nearest-neighbor token mapping does not perfectly preserve the optimized embedding direction. To quantify this effect, we ran the shadow-model optimization on 500 randomly generated strings and compared the optimized continuous embedding with the embedding of the reconstructed discrete tokens. The results are shown below.
>
> | Metric            | Mean | Std  |
> |-------------------|------|------|
> | Cosine similarity | 0.81 | 0.06 |
>
> The cosine similarity of 0.81 indicates that the reconstructed text preserves most, but not all, of the optimized signal. This fidelity loss weakens the effectiveness of the shadow-model approach and explains why its promotion performance is consistently lower than the query-based solution, which avoids the continuous-to-discrete reconstruction step entirely.

---

### Author Response · Authors · 2025-12-02
**Rebuttal Summary**

We thank all reviewers for their constructive feedback.

### Paper Summary

LLM-based search engines generate ranked recommendations that are strongly constrained by the initial retrieval order, which can disadvantage small businesses or creators whose items appear low in the retrieved list. To address this, we propose CORE, the first framework for controlling output rankings in generative search. CORE provides two approaches: a shadow-model optimizer that performs gradient-based updates using a surrogate model, and a query-based optimizer that improves item content through an iterative generator–optimizer loop without accessing model internals. Using AmazonCOREBench, we evaluate CORE across four commercial LLMs and show that it consistently improves the ranking of disadvantaged target items.

### Rebuttal Summary

For reviewer **kqGS**, we addressed concerns regarding Generator Optimizer model alignment, hyperparameter sensitivity $\tau$, detectability, defense mechanisms, API cost, transferability beyond product search, shadow model optimization, and retrieval order effects. We provided extensive new experiments, including cross-model alignment studies, $\tau$ efficiency trade-off analysis, defense evaluations, new ranking domains such as MS MARCO and travel itineraries, and additional theoretical clarification. These results demonstrate that CORE remains effective across different alignment conditions and domains. We believe that all concerns have been fully addressed with experiments.

For reviewer **ppC2**, we clarified that all three concerns raised appear to be based on misunderstandings of the problem formulation and the contents already stated in the paper. First, the reviewer stated that “users do not know which item they want to promote,” but our introduction clearly specifies in lines 78-80 that CORE is designed for small businesses and independent creators who modify only their own product metadata when their items are buried in retrieval results. This directly contradicts the reviewer’s assumption. Second, the reviewer evaluated our method under a relevance-based objective, whereas our paper states that the goal is to increase the visibility of a disadvantaged target item that appears at the bottom of the retrieved list and may otherwise remain invisible in the final recommendations. This makes relevance outside the scope of our problem formulation, which is stated in lines 76-80 and revisited in Sections 3.1 and 3.2. Third, the reviewer concluded that “the optimized outputs have worse fluency”, whereas Table 3 and Table 4 show that only the string-based baseline is intentionally unnatural, and both the reasoning-based and review-based strategies maintain low perplexity, natural fluency, and near baseline human detectability while achieving strong promotion success.

For reviewer **Zxr7**, we provided detailed justification of the novelty and the problem formulation, added theoretical analysis in Appendix C, and conducted new cross-domain experiments to confirm that CORE generalizes beyond product search. We also strengthened the connections to prior work and highlighted that existing jailbreak methods, SEO or GEO approaches, and injection-based techniques cannot achieve the ranking control capabilities demonstrated by CORE.

For reviewer **8QWw**, we added theoretical explanation for Eq. 2, clarified the discrete reconstruction step, analyzed shadow-model fidelity with both theoretical bounds and empirical failure cases, added stability analysis over 3000 optimization iterations, and incorporated three potential defenses (perplexity, pattern-based, length-based). We also provided design recommendations for robust generative-ranking systems.

### Initial Review Outcomes

Reviewer **kqGS** gave a positive score (score 6). Reviewer **ppC2** gave a score of 4, but all concerns are based on misunderstandings that contradict statements made in the paper. Because the comments do not apply to the problem actually studied, we believe this review should be discarded. Reviewer **Zxr7** gave a score of 4. Reviewer **8QWw** gave an initial score of 4. The reviewer asked us to address the listed weaknesses, and these weaknesses have been fully resolved in our rebuttal. We expect a favorable reassessment, although the reviewer currently not have had the opportunity to update the score yet.

---

### Meta-Review · Area_Chair_s1Wh · 2026-01-11

**Summary:**

This paper proposes CORE, an optimization method to control output rankings in LLM-based generative search engines. It aims to address the issue that small businesses’ products are underrepresented due to initial retrieval order bias. It verifies effectiveness via two optimization approaches and the AmazonCOREBench benchmark across four LLMs.

The reviewers' key concerns lie in: limited novelty, hyperparameter fragility, high detectability of reasoning strategy, lack of defense discussion, insufficient method explanation, etc. Besides, there is extensive related work on optimization-based attacks for LLMs or traditional deep learning models. Such work shares similar technical soluations, which should be well discussed.

**Reviewer Concerns:**

Addressed Concerns
- Reliance on model alignment and hyperparameter fragility (supported by cross-model experiments and hp analysis)

- High detectability of the reasoning strategy and lack of defense discussion

- High API cost and limited domain scope
etc

Outstanding Concerns
- Limited novelty and lack of theoretical insights. Some SOTA optimization-based attack methods should be well discussed and compared.
- Long-term robustness with continuous LLM updates

**Reviewer Scores:**

- Reviewer kqGS: Initial score 6, expected to remain stable as most concerns are addressed.
- Reviewer ppC2: Initial score 4, likely to increase scores as most concerns are based on misunderstandings of the paper.
- Reviewer Zxr7: Initial score 4, likely to increase after rebuttal due to supplemented theoretical insights and cross-domain generalization experiments.
- Reviewer 8QWw: Initial score 4, expected to increase after rebuttal as most raised weaknesses and questions are resolved.

---

### Decision · Program_Chairs · 2026-01-26

Reject